# Who Is at Higher Risk of SARS-CoV-2 Reinfection? Results from a Northern Region of Italy

**DOI:** 10.3390/vaccines10111885

**Published:** 2022-11-08

**Authors:** Maria Francesca Piazza, Daniela Amicizia, Francesca Marchini, Matteo Astengo, Federico Grammatico, Alberto Battaglini, Camilla Sticchi, Chiara Paganino, Rosa Lavieri, Giovanni Battista Andreoli, Andrea Orsi, Giancarlo Icardi, Filippo Ansaldi

**Affiliations:** 1Regional Health Agency of Liguria (ALiSa), 16121 Genoa, Italy; 2Department of Health Sciences (DiSSal), University of Genoa, 16132 Genoa, Italy; 3Hygiene Unit, San Martino Policlinico Hospital, IRCCS for Oncology and Neurosciences, 16132 Genoa, Italy

**Keywords:** COVID-19, SARS-CoV-2, epidemiological burden, comorbidities, risk factor, vaccination, vaccine

## Abstract

The SARS-CoV-2 pandemic continues to spread worldwide, generating a high impact on healthcare systems. The aim of the study was to examine the epidemiological burden of SARS-CoV-2 reinfections and to identify potential related risk factors. A retrospective observational study was conducted in Liguria Region, combining data from National Vaccines Registry and Regional Chronic Condition Data Warehouse. In the study period (September 2021 to May 2022), 335,117 cases of SARS-CoV-2 infection were recorded in Liguria, of which 15,715 were reinfected once. During the Omicron phase (which predominated from 3 January 2022), the risk of reinfection was 4.89 times higher (*p* < 0.001) than during the Delta phase. Unvaccinated and vaccinated individuals with at least one dose for more than 120 days were at increased risk of reinfection compared with vaccinated individuals with at least one dose for ≤120 days, respectively (odds ratio (OR) of 1.26, *p* < 0.001; OR of 1.18, *p* < 0.001). Healthcare workers were more than twice as likely to be reinfected than non-healthcare workers (OR of 2.38, *p* < 0.001). Lower ORs were seen among people aged 60 to 79 years. Two doses or more of vaccination were found to be protective against the risk of reinfection rather than a single dose (mRNA vaccines: OR of 0.06, *p* < 0.0001, and OR of 0.1, *p* < 0.0001; vector vaccines: OR of 0.05, *p* < 0.0001). Patients with chronic renal failure, cardiovascular disease, bronchopneumopathy, neuropathy and autoimmune diseases were at increased risk of reinfection (OR of 1.38, *p* = 0.0003; OR of 1.09, *p* < 0.0296; OR of 1.14, *p* = 0.0056; OR of 1.78, *p* < 0.0001; OR of 1.18, *p* = 0.0205). Estimating the epidemiological burden of SARS-CoV-2 reinfections and the role played by risk factors in reinfections is relevant for identifying risk-based preventive strategies in a pandemic context characterized by a high circulation of the virus and a high rate of pathogen mutations.

## 1. Introduction

The COVID-19 pandemic continues to spread worldwide and has caused 532,887,351 confirmed cases, including 6,307,021 deaths, as of 7 June 2022 [1].

SARS-CoV-2, an ever-mutating ribonucleic acid (RNA), has been reported to have a heterogeneous genetic composition in different geographic locations [2], increasing the risk of reinfection. The first case of COVID-19 reinfection was reported in Hong Kong in August 2020 [3]. Since then, the emergence of re-infected cases has been reported in many other countries, including the United States [4] and Italy [5].

In particular, in Italy, since August 2021, several cases of COVID-19 reinfections have been recorded more than 90 days (or more than 60 days in cases with genotyping results indicated different variants) after the previous laboratory-confirmed infection with molecular or antigen testing, according to the case definition of the Ministry of Health [6].

In addition, during the same period, Italy experienced a predominant circulation of two different SARS-CoV-2 variants of concern (VOCs) [7]: from 24 August 2021 to 5 December 2021, the Delta variant predominated, and from 3 January 2022, the Omicron variant became prevalent, with an intermediate period in which the epidemic variants shifted from Delta to Omicron (from 6 December 2021 to 2 January 2022) [8].

According to the latest report from the Italian National Institute of Health (Istituto Superiore di Sanità), 519,603 cases of reinfection were reported in Italy from 24 August 2021 to 5 June 2022 (4.0% of all notified cases). However, the incidence of reinfection, which remained stable at around 1% until 6 December 2021 (the reference date for the start of the Omicron variant), increased sharply to 3% in early January 2022, remained stable until the end of March 2022, and further increased between 6 April and 25 May 2022, reaching an incidence of 4.1% and 6.5%, respectively [9]. In particular, the risk of reinfection starting from 3 January 2022 (starting point of the Omicron variant) highlighted an increase in the adjusted relative risk of reinfection (values significantly greater than 1) compared with the previous phases [9]. Omicron variants are characterized by higher contagiousness and an increased incidence of breakthrough infection and reinfection due to enhanced escape mechanisms of neutralizing antibodies.

Meanwhile, despite the concerns about the infectiousness of Omicron variants, such as the latest BA.5, the effectiveness of the COVID-19 vaccine in preventing severe disease remains high, reaching approximately 68% among those vaccinated with a complete cycle for less than 90 days and 87% among booster-vaccinated [9].

In light of the large number of infected and reinfected individuals, and high vaccination coverage, the healthcare burden and deaths are limited; to the best of our knowledge, the scientific literature about the characteristics of the reinfected individuals and associated potential risk factors is limited.

Knowing the frequency and natural course of reinfection is important to develop control/mitigation strategies against SARS-CoV-2 and to better address the preventive measures and the vaccination strategies. According to the latest report from the National Institute of Health in Italy (Istituto Superiore di Sanità), a higher risk was described in women, younger persons (12–49 years), healthcare workers, unvaccinated people and those who have been vaccinated for more than 120 days [9].

Therefore, the objective of this study is to examine the epidemiological burden of SARS-CoV-2 reinfection in Liguria Region from September 2021 to May 2022, analyzing the potential risk factors of reinfected patients. 

## 2. Materials and Methods

### 2.1. Study Design and Timeline

A retrospective observational study was conducted to assess the epidemiological burden of SARS-CoV-2 reinfections and potential associated risk factors in Liguria Region (Italy). First-time infections and reinfections from September 2021 to May 2022 were included in the analysis with data on infections extracted on 1 June 2022.

### 2.2. Study Population 

Patients were captured through the regional administrative flows selecting reinfections according to the Ministry of Health case definition [6]. Specifically, the inclusion criteria considered individuals with a single episode of reinfection, excluding those who had reported more than one. Data were collected taking into account the age group (0–19, 20–39, 40–59, 60–79 and ≥80 years), vaccination status, VOC predominant phase (Delta phase, Transition phase, Omicron phase), gender (male/female), healthcare worker status (yes/no), nationality (Italian/non-Italian) and comorbidities. Individuals with an interval of 14 days after vaccination were considered as fully vaccinated; those who tested positive for SARS-CoV-2 within 14 days after vaccination were considered as unvaccinated; those who tested positive within 14 days of the second vaccination were classified as having received a single dose. Participants were classified as unvaccinated (never vaccinated or 0–14 days from first dose), vaccinated (at least one dose) within 120 days or vaccinated (at least one dose) for more than 120 days. Regarding the analysis of the vaccination status, vaccine formulations were classified into four groups: mRNA vaccines, which included BNT162b2 and mRNA-1273 vaccines; viral vector vaccines (ChAdOx1 and Ad26.COV2.S); protein vaccines (NVX-CoV2373); and mixed vaccines, which included the administration of a first dose with a viral vector vaccine and subsequent doses with an mRNA vaccine. Vaccinated individuals were divided into single-dose vaccinated, primary-cycle vaccinated (administration of two doses) and vaccinated with booster, additional or second booster doses. Number of days between participants’ study initiation (1 September 2021) and positive SARS-CoV-2 PCR/antigen test results were used to calculate person–time and days to infection and reinfection [10].

### 2.3. Data Sources and Statistical Analysis

Different data sources were used: National Vaccine Registry and regional administrative flows, such as Chronic Condition Data Warehouse and regional Data Warehouse for vaccination status, comorbidities, demographics and subjects positive to SARS-CoV-2, respectively. Odds ratios (ORs) for SARS-CoV-2 reinfections were estimated with the corresponding 95% confidence intervals (CIs). Multivariate logistic regression was used to calculate odds ratios to evaluate the potential risk factors for reinfection compared with the first infection. Data were analyzed using JMP version 13.0.0 software (SAS Institute, Cary, NC, USA). A value of *p* < 0.05 was considered statistically significant.

## 3. Results

### Epidemiological and Clinical Burden

During the study period, on a total Ligurian population of 1,545,174 individuals, 335,117 cases of SARS-CoV-2 infection were recorded in Liguria, of which were 15,795 reinfections (15,715 individuals with one reinfection), indicating an overall reinfection rate of 4.95 per hundred first infections. The first recorded SARS-CoV-2 reinfection case was in early September 2021 (Week 35). Since then, from Week 2021-51, the number of SARS-CoV-2 reinfections markedly increased, probably due to both the increase in the number of first infections and in accordance with the emergence of the more infectious Omicron variant, reaching a peak in Week 2022-03. Thereafter, a rapid decline in first infections, reinfections and deaths until Week 2022-10, when there was a gradual increase in reinfections and conversely a decrease in early infections and deaths, was observed (Figure 1).

In particular, the risk of reinfection during the Omicron phase was 4.89 times higher (95% CI of 4.19–5.72, *p* < 0.001) than during the Delta phase.

Regardless of the predominant variant, the risk of reinfection was 1.26 (95% CI of 1.22–1.31, *p* < 0.001) times higher in unvaccinated individuals and 1.18 (95% CI of 1.13–1.23, *p* < 0.001) times higher in individuals vaccinated with at least one dose for more than 120 days than in individuals vaccinated with at least one dose for ≤120 days. 

The median age of COVID-19-reinfected individuals was 41 years (interquartile range: 23–53). Furthermore, female individuals showed a 17% higher risk of reinfection than the male gender (OR of 1.17, 95% CI of 1.13–1.21, *p* < 0.0001). Non-Italian individuals had a 15% higher risk of reinfection than Italian residents (OR of 1.15, 95% CI of 1.09–1.21, *p* < 0.001); however, after data adjustment, the results showed no associations with a higher risk of reinfection. Healthcare workers were more than twice as likely to be reinfected than non-healthcare workers (OR of 2.38, 95% CI of 2.25–2.52, *p* < 0.001). Individuals aged between 60 and 79 years old had significantly lower odds of becoming reinfected than the other age groups (Table 1).

With regard to vaccination status, the incidence rate of SARS-CoV-2 infection in vaccinated individuals with a single dose was 66.64 per 10,000 person days (PDs), 69.82 in the primary-cycle vaccinated and 49.07 in the booster/additional/second booster group. For reinfected individuals, an incidence rate of reinfection equal to 20.35 (per 10,000 PDs) was detected in vaccinated individuals with a single dose compared with 1.96 and 1.87 (per 10,000 PDs) in vaccinated individuals with primary cycle and subsequent doses, respectively. Furthermore, primary cycle and subsequent doses resulted protective factors in preventing reinfections rather than a single-dose vaccination (mRNA vaccines: OR of 0.06, 95% CI of 0.06–0.07, *p* < 0.0001; and OR of 0.1, 95% CI of 0.09–0.11, *p* < 0.0001; vector vaccines: OR of 0.05, 95% CI of 0.04-0.07, *p* < 0.0001) (Table 2). For mixed vaccines, at least three doses resulted more protective against reinfection rather than only two doses. The small number of subjects vaccinated with protein vaccines did not allow us to have reliable results, showing no statistically significant differences in the number of doses administered.

Relatively to the clinical status detected in May 2022, among the 15,715 reinfected individuals, 12,778 (81.31%) were recovered; a total of 1237 (7.87%) were asymptomatic, and 1172 (7.46%) were pauci-symptomatic; a total of 425 (2.70%) had a mild clinical status, and 14 (0.09%) had a severe or critic clinical status, while 89 (0.57%) were dead.

Most of reinfected individuals (12,843; 81.72%) were healthy, while 2872 (18.28%) had at least one comorbidity. Among reinfected individuals, persons aged 60 years and older had a seven-fold higher risk to have at least one underlying chronic disease than other age groups (OR of 7.39, 95% CI of 6.73–8.14, *p* < 0.0001); chronic renal failure (OR of 1.38, 95% CI of 1.16–1.65, *p* = 0.0003), cardiovascular diseases (OR of 1.09, 95% CI of 1.01–1.19, *p* = 0.0296), bronchopneumopathy (OR of 1.14, 95% CI of 1.04–1.26, *p* = 0.0056), neuropathy (OR of 1.78, 95% CI of 1.58–2.01, *p* < 0.0001) and autoimmune diseases (OR of 1.18, 95% CI of 1.03 -1.37, *p* = 0.0205) were the most implicated comorbidities in patients with reinfection compared with non-reinfected individuals (Table 3). 

In addition, an in-depth analysis of comorbidities revealed that among patients with chronic renal failure, those undergoing dialysis had an almost 3 times higher risk of reinfection (OR of 2.77, 95% CI of 1.76–4.38, *p* < 0.0001). Among the chronic cardiovascular diseases, heart failure and cerebral vasculopathy were the most involved in the risk of reinfection, with 1.24 (95% CI of 1.04–1.47, *p* = 0.0184) and 1.49 (95% CI of 1.29–1.71, *p* < 0.0001) times higher risk than non-reinfected individuals.

Patients with asthma and respiratory failure/oxygen therapy showed 1.17-fold (95% CI of 1.05–1.33, *p* = 0.0070) and 1.67-fold (95% CI of 1.23–2.25, *p* = 0.0009) increased risk of reinfection, respectively, compared with non-reinfected individuals.

Among patients with neuropathy, the risk of reinfection was almost twice as high for those suffering from epilepsy (OR of 1.47, 95% CI of 1.18–1.83, *p* = 0.0006), Parkinson’s and Parkinsonism disease (OR of 1.54, 95% CI of 1.16–2.04, *p* = 0.0029) and Alzheimer’s disease (OR of 1.44, 95% CI of 1.07–1.93, *p* = 0.0159). The risk was about four times higher for people with dementia (OR of 3.71, 95% CI of 3.04–4.52, *p* < 0.0001). Finally, in individuals suffering from autoimmune disease, Hashimoto’s thyroiditis was the most implicated disease in reinfected individuals (OR of 1.19, 95% CI of 1.00–1.44, *p* = 0.0479) (Table 4).

Overall, the death rate was 7.14 per 1000 first infections (2280 dead individuals without reinfection) and 5.66 per 1000 reinfections (89 dead individuals with reinfection).

Among individuals who died of SARS-CoV-2, reinfected individuals had a two-fold higher risk of having neuropathy than non-reinfected individuals (OR of 2.31, 95% CI of 1.43–3.75, *p* = 0.0007), with a four times higher risk for people with dementia (OR of 3.57, 95% CI of 2.02–6.30, *p* < 0.0001). After adjusting data by age and gender, neuropathy and dementia resulted not to be associated with a higher risk of reinfection (Table 5).

## 4. Discussion

The impact of the immune system memory developed after SARS-CoV-2 infection and reinfection is an object of interest for the scientific community. Immune memory to SARS-CoV-2 can be generated by natural infection, vaccination or hybrid immunity (combination of infection-induced immunity and vaccine-induced immunity) and can provide sustained protection against disease by means T-cell memory, B-cell memory and long-lasting antibody responses [11]. Evidence from large observational studies in healthcare workers and the general population suggests that SARS-CoV2-immunity post-infection confers a level of protection against COVID-19 [12,13,14].

The risk of reinfection depends on immune status, infection severity, cross-immunity, age and other immunological factors such as T-cell and B-cell memory or lack of antibody neutralizing capacity [15].

Our study results showed that SARS-CoV-2 reinfection had a similar trend to that observed for primary infection, showing an increasing trend with the spread of the Omicron variant. In particular, the risk of reinfection during the Omicron phase was found to be 4.89 times higher compared with during the Delta phase. Despite the marked increase in the reinfection rates during the Omicron wave, the deaths decreased and remained close to zero values. Our findings are consistent with other studies suggesting that the Omicron variant may evade prior immunity through infection or vaccination but less frequently causes severe clinical outcomes [16,17,18,19]. Indeed, the effect of vaccination on the reduction in the death risk was found in several studies during the Omicron wave [20,21].

Reinfection appears to occur more frequently among women. As some authors reported, the highest risk among women seems to be due to their greater exposure to the infectious disease in work activities, especially in occupations that require proximity to other individuals, such as in the educational or medical fields [22,23]. 

According to our findings, healthcare workers are at an increased risk of SARS-CoV-2 reinfection, which could be plausible given the high risk of front-line work exposure to COVID-19 among these individuals [24,25,26,27,28]. Indeed, healthcare workers are heavily exposed to SARS-CoV-2 during their work. Although personal protective equipment such as masks protect healthcare workers from viral infections, they may not always protect them due to improper donning and the removal of previously contaminated masks if proper hand hygiene and disinfection are not performed [29]. Thus, the lack of adequate infection control measures represents another potential reason for protective failures and nosocomial transmission. Personal protective equipment and hand hygiene have been shown to be essential in the protection from respiratory viruses, and hand hygiene is the primary recommended measure by the WHO to control cross-infection [30,31], but the frequency of emergency procedures performed in healthcare facilities during the COVID-19 pandemic was high, causing a minimized use of these infection control measures and decreased risk perception [29]. The current COVID-19 pandemic has strained healthcare workers around the world affecting their daily work. Healthcare workers, especially nurses, are typically the professional group that spends the most time in direct contact with patients compared with other healthcare workers, and it is essential to maintain the quality of their working lives in order for quality care and good patient outcomes to be achieved [32]. 

Thus, national health authorities should support or increase manpower, especially by recruiting new healthcare workers, providing appropriate personal protective equipment kits during treatment or exposure to infected patients, minimizing work pressure, highly motivating healthcare workers and recognizing the responsibilities of different healthcare professions [33].

Immunization has shown a protective effect against reinfection, especially among those vaccinated for less than 120 days. This effect can be explained by the high vaccination coverage achieved in Italy and by the effectiveness of the vaccine in preventing the onset of SARS-CoV-2 infection and in preventing cases of severe disease [34].

Furthermore, the risk of COVID-19 reinfection was found to be significantly lower in those vaccinated after the primary cycle or subsequent doses than in those vaccinated with a single dose, confirming the need for a complete vaccination cycle whenever possible, also taking into account the duration of immunity against COVID-19 infection, an important aspect that is not yet fully understood [35], raising the risk of COVID-19 re-infection.

Our data confirm the robustness of the applied vaccination strategies that recommend that one of the objectives of COVID-19 vaccination campaigns continues to be to protect health systems, reducing COVID-19 hospitalization, severe disease and death. Improving vaccine uptake in eligible individuals who are yet to receive them remains a priority, especially for population groups at higher risk of severe outcomes [36].

The evaluation of the risk of reinfection by age group showed a lower reinfection risk in individuals aged 60–79 years; this is probably linked to their behaviors. Indeed, the association between age and the adoption of preventive measures has been examined in other studies, and it has shown inconsistencies across different countries and during different pandemics [37,38,39,40]. However, some authors have attributed hygiene practices and social distancing to greater compliance with government regulations and greater fear of infection due to greater vulnerability among those aged 65 years and older than among younger people aged 18–34 years [37,41]. The current study also revealed that 18.28% of the individuals had one or more comorbidities and that the most implicated in reinfection were chronic renal failure, cardiovascular diseases, bronchopneumopathy, neuropathy and autoimmune diseases, especially in individuals on dialysis and with heart failure, cerebrovascular disease, asthma, respiratory failure/oxygen therapy, epilepsy, Parkinson’s and Parkinsonism disease, Alzheimer’s disease, dementia and Hashimoto’s thyroiditis. It is worth noting that dialysis patients, being dependent on regular treatment sessions, often using public transportation, expose themselves to the risk of community-transmitted infection and have contacts with healthcare workers, potential sources of transmission. These patients are at a higher risk of worsened prognoses with COVID-19, especially those with additional comorbidities, and are particularly susceptible to SARS-CoV-2 infection, as treatment necessitates frequent visits to outpatient dialysis units [42,43].

Although several scientific studies have reported an association between certain comorbidities and the development of severe forms of SARS-CoV-2 infection [44,45,46,47,48,49,50], literature studies on the association between comorbidities and the risk of reinfection are limited.

A systematic review published in 2022, reporting data from 1 December 2019 to 1 September 2021, described that hypertension and obesity were the most common among reinfected patients, followed by end-stage renal failure, asthma, chronic obstructive pulmonary disease (COPD), dementia, dyslipidemia and type 2 diabetes [51]. 

Other studies have also reported a higher risk of reinfection in individuals with end-stage renal failure, hypertension, diabetes, chronic respiratory disease, liver disease and a history of cardiovascular disease [52,53,54,55].

To our knowledge, this is one of the first literature studies to assess risk factors in SARS-CoV-2-reinfected individuals considering a large subset; the epidemiological burden of SARS-CoV-2 reinfections and the role played by risk factors for reinfections have immediate implications for public health policy, and risk-based prevention strategy identification is an appropriate and necessary aspect to consider.

The present study has some limitations. Although healthcare workers, especially those in high-risk occupations, are highly responsible and routinely screened, and the reinfection rate may not be very high, the large number of asymptomatic, undetected primary infected patients may underestimate the reinfection rate.

Thus, reinfection is more difficult to document when asymptomatic cases of reinfections are included, and precisely for this reason, it is essential that countries continuously monitor rates and patterns of reinfection to guide policy choices to strengthen data management systems.

It is, therefore, crucial to maintain the microbiological and epidemiological surveillance of COVID-19 [56], which continuously and systematically collects, compares, and analyzes information on all cases of SARS-CoV-2 infection confirmed by molecular and antigen diagnosis at regional reference laboratories in Italy. This is a necessary and useful observational tool both to inform citizens about the impact and evolution of the epidemic and to support decision-making in the public health response of health authorities.

Furthermore, as the SARS-CoV-2 virus continues to evolve, VOCs may increase transmissibility and reduce the effectiveness of COVID-19 vaccines. Therefore, laboratories around the world need to continue genotyping SARS-CoV-2 variants.

To summarize the study results, recurrent and severe SARS-CoV-2 infections tend to occur more frequently in unvaccinated individuals, those with weak immunity, and those with pre-existing health conditions, who have reduced access to services and quality healthcare and who live and work in environments that increase their risk of infection. As COVID-19 revealed significant differences in outcomes among high-risk populations, pandemics also provide an opportunity for policymakers to take steps to reduce inequalities in the long term based on the latest findings. Furthermore, our findings suggest that health systems must focus on community interventions to prevent the spread of infection to vulnerable populations, especially high-risk groups, and the need to maintain epidemiological and virological surveillance for policymaking and response activities to prevent the impact on health services. 

## 5. Conclusions

Considering the different epidemic phases, the study results showed a five times higher risk of reinfection during the Omicron phase compared with the circulation phase of the Delta variant. Regardless of the predominant variant, being unvaccinated was the most relevant risk factor for reinfection. Additionally, healthcare workers showed a two-fold higher risk of SARS-CoV-2 reinfection rather than non-healthcare workers.

It is important to underline the weight of some comorbidities in individuals with reinfection, such as chronic renal failure, cardiovascular diseases, bronchopneumopathy, gastroenteropathy, neuropathy and autoimmune diseases.

These results are relevant for identifying the best prevention strategies in a pandemic context characterized by a high circulation of the SARS-CoV-2 virus and a high rate of pathogen mutations.

## Figures and Tables

**Figure 1 vaccines-10-01885-f001:**
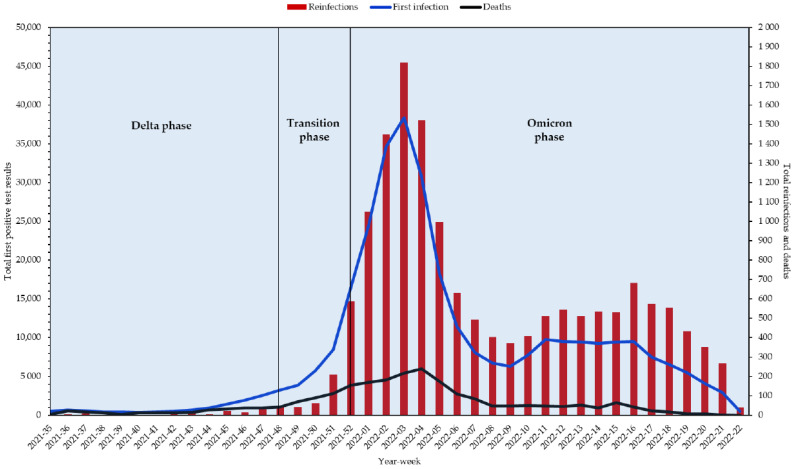
Weekly number of SARS-CoV-2 reinfections, primary infection and deaths in Liguria Region.

**Table 1 vaccines-10-01885-t001:** Main characteristics and odds ratios of SARS-CoV-2-reinfected individuals in Liguria.

Variable		Reinfection (15,715)	PDs (53,891,505)	Incidence Per 10,000 PD	Unadjusted OR (95% CI)	*p*-Value	Adjusted OR (95% CI)	*p*-Value
Epidemic phase	Delta	161	958,178	1.68	Reference		Reference	
	Transition	903	4,013,487	2.25	2.36 (1.99–2.79)	<0.001	2.40 (2.03–2.84)	<0.001
	Omicron	14,651	48,919,840	2.99	4.89 (4.19–5.72)	<0.001	5.27 (4.51–6.17)	<0.001
Vaccination Status	Vaccinated with at least one dose for ≤120 days	5350	22,330,172	2.40	Reference		Reference	
	Vaccinated with at least one dose for >120 days	4581	14,764,645	3.10	1.18 (1.13–1.23)	<0.001	1.29 (1.25–1.35)	<0.001
	Unvaccinated	5784	16,796,688	3.44	1.26 (1.22–1.31)	<0.001	1.62 (1.55–1.69)	<0.001
Gender	Male	6812	25,089,342	2.72	Reference		Reference	
	Female	8903	28,802,163	3.09	1.17 (1.13–1.21)	<0.001	1.11 (1.08–1.15)	<0.001
Age-group	0–19	3235	12,177,126	2.66	1.37 (1.29–1.46)	<0.001	1.22 (1.15–1.30)	<0.001
	20–39	4293	12,222,708	3.51	1.84 (1.74–1.96)	<0.001	1.76 (1.66–1.87)	<0.001
	40–59	5804	17,582,644	3.30	1.77 (1.68–1.88)	<0.001	1.66 (1.57–1.76)	<0.001
	60–79	1588	8,776,040	1.81	Reference		Reference	
	≥80	795	3,132,987	2.54	1.46 (1.34–1.60)	<0.001	1.52 (1.39–1.66)	<0.001
Nationality	Italian	14,080	304,189	462.87	Reference		Reference	
	Non-Italian	1635	30,928	528.65	1.15 (1.09–1.21)	<0.001	1.04 (0.99–1.10)	0.1016
Healthcare worker	No	14,265	51,391,133	2.78	Reference		Reference	
	Yes	1450	2,500,372	5.80	2.38 (2.25–2.52)	<0.001	2.43 (2.29–2.58)	<0.001

CI, confidence interval; OR, odds ratio; PDs, person days. Adjusted ORs include all factors in the table, mutually adjusted for each other.

**Table 2 vaccines-10-01885-t002:** Vaccinated individuals and odds ratios of SARS-CoV-2-reinfected individuals in Liguria.

Variable *		Overall (335,037)	Not Reinfected (319,322)	Reinfected (15,715)	PDs (37,094,817)	OR (95% CI)	*p*-Value
All vaccines	Single dose	10,216 (66.64)	7096 (46.29)	3120 (20.35)	1,533,000	Reference	
	Primary cycle	118,547 (69.82)	115,218 (67.86)	3329 (1.96)	16,979,202	0.07 (0.06–0.07)	<0.0001
	Booster/additional dose/second booster	94,675 (49.07)	91,193 (49.07)	3482(1.87)	18,582,615	0.09 (0.08–0.09)	<0.0001
mRNA	Single dose	9930 (66.58)	6886 (46.17)	3044 (20.41)	1,491,509	Reference	
	Primary cycle	103,560 (68.72)	100,680 (66.81)	2880 (1.91)	15,070,369	0.06 (0.06–0.07)	<0.0001
	Booster/additional dose/second booster	75,953 (50.82)	72,715 (48.65)	3238 (2.17)	14,946,436	0.1 (0.09–0.11)	<0.0001
Vector	Single dose	243 (75.38)	170 (52.74)	73 (22.65)	32,235	Reference	
	Primary cycle	9123 (79.74)	8922 (77.98)	201 (1.76)	1,144,134	0.05 (0.04–0.07)	<0.0001
Protein	Single dose	43 (46.46)	40 (43.22)	3 (3.24)	9256	Reference	
	Primary cycle	32 (41.31)	31 (40.02)	1 (1.29)	7747	0.43 (0.04–4.34)	0.4743
Mixed	Primary cycle	5832 (77.05)	5585 (73.78)	247 (3.26)	756,952	Reference	
	Booster/additional dose/second booster	18,722 (51.49)	18,478 (50.82)	244 (0.67)	3636,179	0.29 (0.25–0.36)	<0.0001

* Expressed as number and incidence rate per 10,000 PDs. CI, confidence interval; OR, odds ratio; PDs, person days.

**Table 3 vaccines-10-01885-t003:** Comorbidities implicated in reinfected SARS-CoV-2 patients from September 2021 to May 2022.

Comorbidity	OR	Lower 95% CI	Upper 95% CI	*p*-Value
02—Transplant	1.06	0.68	1.67	0.7872
03—Chronic renal failure	1.38	1.16	1.65	0.0003
04—HIV/AIDS	0.52	0.27	1.00	0.0503
05—Cancer	0.95	0.88	1.04	0.2785
06—Diabetes	0.91	0.81	1.01	0.0877
07—Cardiovascular disease	1.09	1.01	1.19	0.0296
08—Bronchopneumopathy	1.14	1.04	1.26	0.0056
09—Gastroenteropathy	1.09	0.98	1.22	0.0941
10—Neuropathy	1.78	1.58	2.01	<0.0001
11—Autoimmune disease	1.18	1.03	1.37	0.0205
12—Endocrine and metabolic disease	0.89	0.82	0.97	0.0103
13—Rare disease	1.05	0.89	1.24	0.5391
14—Psychosis	1.28	1.04	1.59	0.0215

**Table 4 vaccines-10-01885-t004:** Detail of the main comorbidities implicated in reinfected SARS-CoV-2 patients from September 2021 to May 2022.

Comorbidity	OR	Lower 95% CI	Upper 95% CI	*p*-Value
03—Chronic renal failure				
03A—Chronic renal failure—dialysis	2.77	1.76	4.38	<0.0001
03B—Chronic renal failure—no dialysis	1.27	1.05	1.54	0.0143
07—Cardiovascular disease				
07S1—Arterial hypertension	0.75	0.70	0.79	<0.0001
07S2—Ischemic heart disease	0.97	0.84	1.12	0.6772
07S3—Valvular heart disease	0.99	0.80	1.24	0.9882
07S4—Arrhythmic myocardiopathy	1.03	0.89	1.19	0.6850
07S5—Non-arrhythmic myocardiopathy	1.08	0.89	1.31	0.4125
07S6—Heart failure	1.24	1.04	1.47	0.0184
07V1—Arterial vasculopathy	1.19	0.94	1.51	0.1575
07V2—Venous vasculopathy	0.99	0.60	1.63	0.9630
07V3—Cerebral vasculopathy	1.49	1.29	1.71	<0.0001
08—Bronchopneumopathy				
08A—Asthma	1.17	1.05	1.33	0.0070
08B—Chronic obstructive pulmonary disease	1.09	0.94	1.25	0.2636
08C—Respiratory failure/oxygen therapy	1.67	1.23	2.25	0.0009
10—Neuropathy				
10A—Epilepsy	1.47	1.18	1.83	0.0006
10B—Parkinson’s and Parkinsonisms disease	1.54	1.16	2.04	0.0029
10C—Alzheimer’s disease	1.44	1.07	1.93	0.0159
10D—Multiple sclerosis	1.02	0.68	1.54	0.9237
10E—Optic neuromyelitis	-	-	-	-
10F—Dementia	3.71	3.04	4.52	<0.0001
11—Autoimmune disease				
11A—Rheumatoid arthritis	1.39	0.97	1.97	0.0698
11B—Systemic lupus erythematosus	0.77	0.29	2.10	0.6151
11C—Systemic sclerosis	1.42	0.66	3.06	0.3674
11D—Sjogren’s disease	1.45	0.63	3.32	0.3780
11E—Ankylosing spondylitis	0.85	0.31	2.30	0.7442
11F—Myasthenia gravis	1.12	0.35	3.61	0.8381
11G—Hashimoto’s thyroiditis	1.19	1.00	1.44	0.0479
11H—Immune hemolytic anemias	0.70	0.09	5.14	0.7265
11I—Psoriasis and psoriatic arthropathy	1.14	0.73	1.80	0.5561

**Table 5 vaccines-10-01885-t005:** Comorbidities implicated in dead reinfected SARS-CoV-2 patients from September 2021 to May 2022.

Comorbidity	Unadjusted OR	Lower 95% CI	Upper 95% CI	*p*-Value	Adjusted OR	Lower 95% CI	Upper 95% CI	*p*-Value
02—Transplant	-	-	-	-	-	-	-	-
03—Chronic renal failure	1.23	0.65	2.35	0.5247	1.17	0.58	2.35	0.6662
04—HIV/AIDS	-	-	-	-	-	-	-	-
05—Cancer	0.80	0.46	1.38	0.4230	0.79	0.45	1.40	0.4320
06—Diabetes	0.93	0.53	1.63	0.7935	0.85	0.47	1.54	0.5899
07—Cardiovascular disease	1.44	0.94	2.21	0.0938	1.43	0.88	2.33	0.1515
08—Bronchopneumopathy	1.15	0.62	2.14	0.6583	1.13	0.59	2.16	0.7085
09—Gastroenteropathy	0.85	0.44	1.61	0.6079	0.79	0.41	1.56	0.5105
10—Neuropathy	2.31	1.43	3.75	0.0007	1.47	0.41	5.18	0.5524
10A—Epilepsy	1.36	0.32	5.71	0.6778	0.75	0.15	3.84	0.7306
10B—Parkinson’s And Parkinsonisms	1.82	0.77	4.29	0.1694	1.28	0.38	4.34	0.6968
10C—Alzheimer’s	1.27	0.50	3.20	0.6109	0.77	0.24	2.51	0.6686
10D—Multiple sclerosis	-	-	-	-	-	-	-	-
10E—Optic neuromyelitis	-	-	-	-	-	-	-	-
10F—Dementia	3.57	2.02	6.30	<0.0001	2.49	0.78	7.99	0.1221
11—Autoimmune disease	1.09	0.26	4.57	0.9039	0.91	0.21	3.99	0.9042
12—Endocrine and metabolic disease	0.93	0.53	1.64	0.8015	0.89	0.49	1.61	0.6983
13—Rare disease	1.51	0.19	11.49	0.6891	1.65	0.21	13.09	0.6359
14—Psychosis	1.43	0.34	6.05	0.6244	0.82	0.18	3.68	0.7988

Adjusted ORs include all factors in the table adjusted for age and gender.

## Data Availability

The data are not publicly available due to privacy or ethical restrictions.

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
