# Peer review of "Who Is at Higher Risk of SARS-CoV-2 Reinfection? Results from a Northern Region of Italy"

_vaccines, 2022, doi:10.3390/vaccines10111885_

Round 1

Reviewer 1 Report (New Reviewer)

Review, Piazza et al., Vaccines, 2022: Who is at higher risk of SARS-CoV-2 reinfection? Results from a Northern Region of Italy

Summary

In the study by Piazza et al, the group proposed to examine the epidemiological burden of SARS-CoV-2 reinfections and to identify potential related risk factors in the Liguria Region.

They observed that higher risk was described in women, younger persons (12-49 years), healthcare workers, unvaccinated people and those who have been vaccinated for more than 120 days. 

Overall, this manuscript is well written and information is pertinent. Nevertheless, some minor comments have to be addressed before publication.

Minor comments

Please note that throughout the manuscript, certain words/sentences are always in red.

1.     Line 26, the authors should precise when the Omicron phase occurs.

2.     Authors should introduce the term “breakthrough infection” when discussing re-infection. This is a commonly used term in the COVID-19 literature.

3.     Line 56, was confirmed infection determined by PCR test? Please, specify.

4.     Is the risk of being re-infected different depending of the circulating VOC?

5.     Line 70-71, the conclusions should be placed after the introduction of the hypothesis and aims.

6.     Line 82, a space is missing between against and SARS-CoV-2.

7.     Figure 1, are the PCR results shown?

8.     Line 178, correct “years and”

9.     The discussion has a lot of redundancy, authors should reduce it for more clarification. Authors should also incorporate the impact of the immune system memory, developed after infection and reinfection and which is more mature and able to protect against new SARS-CoV-2 infection.

Author Response

Reviewer 1

Review, Piazza et al., Vaccines, 2022: Who is at higher risk of SARS-CoV-2 reinfection? Results from a Northern Region of Italy

Summary

In the study by Piazza et al, the group proposed to examine the epidemiological burden of SARS-CoV-2 reinfections and to identify potential related risk factors in the Liguria Region.

They observed that higher risk was described in women, younger persons (12-49 years), healthcare workers, unvaccinated people and those who have been vaccinated for more than 120 days. 

Overall, this manuscript is well written and information is pertinent. Nevertheless, some minor comments have to be addressed before publication.

Minor comments

Please note that throughout the manuscript, certain words/sentences are always in red.

Thank you for your comment, we modified all the text.

  1. Line 26, the authors should precise when the Omicron phase occurs.

Thank you for your comment, we added this information.

  1. Authors should introduce the term “breakthrough infection” when discussing re-infection. This is a commonly used term in the COVID-19 literature.

Thank you for your comment. We added the term as suggested.

  1. Line 56, was confirmed infection determined by PCR test? Please, specify.

We thank you for your comment. We have replaced the reference with the correct one and better specified this concept in the text.

  1. Is the risk of being re-infected different depending of the circulating VOC?

We thank you for your comment. As reported by the report from the Istituto Superiore di Sanità the risk of being re-infected seems to depend to the circulating VOC. Indeed, the risk of reinfection starting from 3 January 2022 (starting point of the Omicron variant) highlighted an increase adjusted relative risk of reinfection (values significantly greater than 1) than the previous phases. We added this sentence in the main text.

  1. Line 70-71, the conclusions should be placed after the introduction of the hypothesis and aims.

Thank you for your comment. We modified as suggested.

  1. Line 82, a space is missing between against and SARS-CoV-2.

Thank you for your comment. We modified as suggested.

  1. Figure 1, are the PCR results shown?

Thank you for your comment. Accordingly with the previous comment we reported that in Italy, since August 2021, several cases of COVID-19 reinfections have been recorded more than 90 days (or more than 60 days in cases with genotyping results indicated different variants) after the previous laboratory-confirmed infection with molecular or antigen testing, according to the case definition of the Ministry of Health.

  1. Line 178, correct “years and”

Thank you for your comment. We modified as suggested.

  1. The discussion has a lot of redundancy, authors should reduce it for more clarification. Authors should also incorporate the impact of the immune system memory, developed after infection and reinfection and which is more mature and able to protect against new SARS-CoV-2 infection.

Thank you for your comment. We have deepened the concepts suggested.

Reviewer 2 Report (New Reviewer)

Review

General comments:

The research utilized the valuable data from Liguria Region. The result of this research can be served as the reference for not only the local government but also the central government of Italy. However, the research still needs to add more information on their research method.

1.     In your methodology, you mentioned you used multivariate logistic regression. Therefore, we assume all the models were adjusted with confounders. However, it does not look like other table (but table 1) included other covariates. Without adjustment of confounders, we cannot make sure the true effect was caused by comorbidities or by other factors such as age, or vaccination status. I suggest at least age and gender should be adjusted.

2.     The study seems to be a cross-sectional study exploring the risk and protective factors towards COVID-19 reinfection. However, you included the information about person time in the research. Please explain how person time was calculated. What is the starting day to count the risk of each person?

3.     Line 101: it should be the individuals with an interval of 14 days after vaccination “with the second dose” were considered…

4.     Line 108: Mrna-1273 should be changed into mRNA-1273

5.     Line 115-117: Please verify what kind of information was retrieved from which data sources

6.     Line 124-126: Please provide the information of total population in Liguria Region.

7.     Line 134-135: The name of the figure should be placed under the figure, not above

8.     Line 151: Were all the variables fitted in one model? Can you elucidate the variable selection method used. Why did you include nationality in the model? Are these foreign residents or those visiting with history of recent travel to the region. Please clarify the inclusion of the variable nationality in the analysis

9.     Line 155-158: Since you first mentioned the information about the total vaccination against infection, this information should be moved to the top of table 2.

10.  Line 158-160: The number differs from what presented in Table 2.

11.  In the stratified analysis (table 3), what is the reference category per stratum, patients with or without comorbidities? What are the reference strata for? What is the objective of the analysis? What is the reference category for comorbidities? I think this analysis is not valuable for the research. I would recommend the authors to consider studying the association between comorbidity and bad outcome (death or critical clinical status).

12.  Table 6, Line 217-220: Did you do any adjustment for age and gender in the model? I believe Neuropathy and dementia associated with higher rate of mortality at old age.

13.  Line 227-228: Is it truly > 5 fold? The number presented in table 1 was only 4.89.

14.  Line 228-230: The conclusion about lower death rate during Omicron phases was based on the observation at Figure 1 only. Since you did not do any adjustment, this effect might be caused by others factors such as vaccination, especially, in recent studies, the effect of vaccination on reduction of death risk was elucidated.

15.  Line 237-251: Base on this paragraph, you assumed the foreigner identity in Ligurian region as disadvantaged population. Do you have data or reference to prove it since the reference you used here belonged to Kuwait and the USA which might not be applicable to this study.

16.  Line 290-292, this explanation might not be valid because you already adjusted for vaccination status in the model. Therefore, you must assume that everyone has the same vaccination status. You can keep the explanation about the risky behaviors.

17.  Since there has been limited biological evidence on association between the health condition and COVID19 infection, the association between comorbidities and infection could also be considered through behavior pathway. For example, in your research, the people with dialysis had higher chance of reinfection. It might be explained by the high frequency of travelling to the hospital of the patients. However, I did not see this discussion in your manuscript.

Author Response

Reviewer 2

General comments:

The research utilized the valuable data from Liguria Region. The result of this research can be served as the reference for not only the local government but also the central government of Italy. However, the research still needs to add more information on their research method.

  1. In your methodology, you mentioned you used multivariate logistic regression. Therefore, we assume all the models were adjusted with confounders. However, it does not look like other table (but table 1) included other covariates. Without adjustment of confounders, we cannot make sure the true effect was caused by comorbidities or by other factors such as age, or vaccination status. I suggest at least age and gender should be adjusted

We agree with you comment. We added in the table the adjusted Odds Ratio and p-values.

  1. The study seems to be a cross-sectional study exploring the risk and protective factors towards COVID-19 reinfection. However, you included the information about person time in the research. Please explain how person time was calculated. What is the starting day to count the risk of each person?

Thank you for your comment. Number of days between participants’ study initiation (01th September 2021) and positive SARS-CoV-2 PCR/antigen test results were used to calculate person-time at risk and days to infection and reinfection [Rivelli A, Fitzpatrick V, Blair C, Copeland K, Richards J. Incidence of COVID-19 reinfection among Midwestern healthcare employees. PLoS One. 2022 Jan 4;17(1):e0262164. doi: 10.1371/journal.pone.0262164. PMID: 34982800; PMCID: PMC8726474]. This was clarified in the main text.

  1. Line 101: it should be the individuals with an interval of 14 days after vaccination “with the second dose” were considered…

Thank you for your comment. We modified the text as required.

  1. Line 108: Mrna-1273 should be changed into mRNA-1273

Thank you for your comment. We modified the text as required.

  1. Line 115-117: Please verify what kind of information was retrieved from which data sources

We agree with your comment and we modified as required.

  1. Line 124-126: Please provide the information of total population in Liguria Region.

Thank you for your comment. We added the information requested.

  1. Line 134-135: The name of the figure should be placed under the figure, not above

Thank you for your comment. We moved the name of the figure as requested.

  1. Line 151: Were all the variables fitted in one model? Can you elucidate the variable selection method used. Why did you include nationality in the model? Are these foreign residents or those visiting with history of recent travel to the region. Please clarify the inclusion of the variable nationality in the analysis

We agree with your comment. We added the nationality variable in the univariate analysis during the covid-19 data collection phase for the pandemic, the ministry of health requested this distinction. Anyway, as suggested in the previous comment we have adjusted the variables for the confounders and the nationality variable loses its statistical significance. Furthermore we modified the term “foreign” in “non-italian” because we do not have the exact information about the residence.

  1. Line 155-158: Since you first mentioned the information about the total vaccination against infection, this information should be moved to the top of table 2.

We agree with your comment and we modified the table as required.

  1. Line 158-160: The number differs from what presented in Table 2.

Thank you for your comment. The incidence rates reported in the text for reinfected individuals are reported in the values in brackets of the third column in the "all vaccines" category.

  1. In the stratified analysis (table 3), what is the reference category per stratum, patients with or without comorbidities? What are the reference strata for? What is the objective of the analysis? What is the reference category for comorbidities? I think this analysis is not valuable for the research. I would recommend the authors to consider studying the association between comorbidity and bad outcome (death or critical clinical status).

We agree to drop the table because it does not add valuable information for the study. This table was inserted in response to a request from a previous reviewer. Furthermore, as suggested we have left the relative table to comorbidities implicated in dead reinfected SARS-COV-2 patients (table 6).

  1. Table 6, Line 217-220: Did you do any adjustment for age and gender in the model? I believe Neuropathy and dementia associated with higher rate of mortality at old age.

Thank you for your comment. We added in the table the adjusted Odds Ratio and p-values.

  1. Line 227-228: Is it truly > 5 fold? The number presented in table 1 was only 4.89.

We agree with your comment and we modified as required.

  1. Line 228-230: The conclusion about lower death rate during Omicron phases was based on the observation at Figure 1 only. Since you did not do any adjustment, this effect might be caused by others factors such as vaccination, especially, in recent studies, the effect of vaccination on reduction of death risk was elucidated.

Thank you for your suggestion. We added a sentence on the effect of vaccination on the reduction of death during Omicron phase.

  1. Line 237-251: Base on this paragraph, you assumed the foreigner identity in Ligurian region as disadvantaged population. Do you have data or reference to prove it since the reference you used here belonged to Kuwait and the USA which might not be applicable to this study.

Thank you for your comment and according to the previous comment we have deleted this part.

  1. Line 290-292, this explanation might not be valid because you already adjusted for vaccination status in the model. Therefore, you must assume that everyone has the same vaccination status. You can keep the explanation about the risky behaviors.

Thank you for your comment. We modified the text accordingly.

  1. Since there has been limited biological evidence on association between the health condition and COVID19 infection, the association between comorbidities and infection could also be considered through behavior pathway. For example, in your research, the people with dialysis had higher chance of reinfection. It might be explained by the high frequency of travelling to the hospital of the patients. However, I did not see this discussion in your manuscript.

Thank you for your comment. We have deepened the concepts suggested.

This manuscript is a resubmission of an earlier submission. The following is a list of the peer review reports and author responses from that submission.

Round 1

Reviewer 1 Report

This study aims to assess the epidemiological burden of SARS-COV-2 reinfections in Liguria from September 2021 to May 2022 by examining the possible risk factors of reinfected patients to better address the preventative measures and immunization approaches.

The main concern with this ms. is that it is more about risk factors for reinfection other than vaccination status, although a simple comparison is included of unvaccinated vs. vaccinated with at least one dose (> 120 days) vs. vaccinated with at least one dose [≤ 120 days], but with little or no attention to which vaccines were used, what their initial standard dose[s] regimen are (1 or 2), and at this point in the pandemic, the number of booster doses received should be part of the analysis. In short, for publication in Vaccines, I would expect a more complex epidemiological analysis primarily of vaccine and vaccination variables with population and co-morbidity differences as secondary hypotheses or observations. The greater focus in this ms. on comorbidities is important but is better suited to other journals.

1-    The ms. is written carelessly and confusingly.

2-    As just one glaring example, on both line 28 of the abstract and lines 132-134 of the text, the wording seems to compare foreigners and Italian healthcare workforce, which conflates 2 different variables and also misstates the difference value. What I think should have been written is that: “female subjects showed a 17% [NOT approximately 2 times!] higher risk of reinfection than male gender (OR 1.17; 95% CI 1.13-1.21, p=<0.0001). Foreign subjects had a 15% higher risk of reinfection than Italian residents (OR 1.15; 95% CI: 1.09-1.21, p=<0.001). Healthcare workers were more than twice as likely to be reinfected as non-healthcare workers (OR 2.38; 95% CI: 2.25-2.52, p=<0.001).”

3-    Keywords are not related to the study, such as vaccination coverage. “Comorbidities” is written twice (lines 36-37). 

4-    It is always better to explain abbreviations the first time it appears in the article, such as RR (line 29) and RNA (line 42).

5-    line 61: Specify the date range and do not refer to “in the last few weeks.”

6-    No data or table is available for results written in lines (141-142).

7-    A description of the clinical and epidemiological characteristics of the participants is not provided in the result section (line 108).

8-    Please display periods of Delta and Omicron variants and the transition between them in figure 1.

9-    The demographic characteristics of the individuals with different comorbidities are not provided in tables, and there is no table showing the comparison analysis while the authors somehow compared them (lines 153-155).

10- For interpreting the results, write the result from the table next to its word instead of lining them side by side (lines 143-147) (lines 165-168).

11- Endocrine and metabolic disease and Psychosis are also significant in table 2, but detailed analysis like other significant comorbidities is not provided in table 3. 

12- Please correct the wording [should the comma in death rate be a decimal point or “1,000” be 100,000? Also not sure why the parenthetical ratios differ from the preceding numbers, both of which appear to be based on 1,000 denominators] in lines 176-177:

“Overall death rate was 2,280 per 1,000 first infections (7.14 per 1,000) and 89 per 1,000 reinfections (5.66 per 1,000).”

13- No other professional category is mentioned in the study besides healthcare workers. Please delete the word “categories” in line 309.

14- Please find a more specific and rational reason or speculation for explaining the increased reinfection among foreigners instead of saying the non-Italian population made up 10 percent of the population (lines 198-200).

15- It would be preferable to use the word “residents”, “patients”,“individuals,” “persons,” or “people” instead of “subjects” when referring to them. One reason is that this is not experimental design research, but also more generally, the term “subject” may be perceived as demeaning.

16- Finally, some of the writing followed source wording too closely. We don't think it is plagiarized, but suggests laziness in the writing.

Reviewer 2 Report

Estimated Authors of the paper "Who is at higher risk of SARS-CoV-2 reinfection? Results from a Northern Region of Italy",

I've read your paper with great interest. In this ecological study, Piazza et al were able to characterize the overall risk for SARS-CoV-2 reinfection among people from the Italian Liguria region, the respective risk factors, as well as the risk factors for deaths following re-infection.

In this sample, the higher risk of reinfections from omicron than from delta, from vaccination whose last dose occurred ≥ 120 days before the contact at risk; of female gender, among foreign-born individuals, and in HCWs was identified and accurately discussed. Several explanations for the estimates were provided, and are quite consistent with the ragionale of this research.

However, before the eventual acceptance of this study, some formal improvements are required, and more precisely:

1) Figure 1 may be misleading when dealing with death rates, as the correspondent axis is not precisely identified ; double check correspondents legend for y axes. Moreover, I guess whether the design may be considered appropriated as for earlier and later stages of the pandemic the appreciation of incidence cases and deaths may be compromized by the striking difference in the figures;

2) scattered across the text, some typos impair the flow of the main text: for example, row 158: p=< ... please fix as p < ... (several occurrences across the main text, please double check)

3) row 165-167: Among patients with neuropathy, the risk of reinfection was almost twice as high for those suffering from epilepsy, Parkinson's and Parkinsonism and Alzheimer's (OR 1.47, 166 95% CI 1.18-1.83, p=0.0006; OR 1.54, 95% CI 1.16-2.04, p=0.0029; OR 1.44, 95% CI 1.07-1.93, p=0.0159). ... Did you mean: "Among patients with neuropathy, the risk of reinfection was almost twice as high for those suffering from epilepsy, Parkinson's DISEASE and Parkinsonism and Alzheimer's DISEASE (OR 1.47, 95% CI 1.18-1.83, p=0.0006; OR 1.54, 95% CI 1.16-2.04, p=0.0029; OR 1.44, 95% CI 1.07-1.93, p=0.0159)"

4) "Overall death rate was 2,280 per 1,000 first infections (7.14 per 1,000) and 89 per 1,000 reinfections (5.66 per 1,000)". Please check the unit of measure.

Round 2

Reviewer 1 Report

The authors did not respond to our main comment that the study hardly focuses on vaccination-related variables, except in a very minimal way, and they certainly did not respond in their cover letter to that comment.  The top of our review was clearly labeled as "The main concern with this ms..." but was ignored. Here again is that comment:

The main concern with this ms. is that it is more about risk factors for reinfection other than vaccination status, although a simple comparison is included of unvaccinated vs. vaccinated with at least one dose (> 120 days) vs. vaccinated with at least one dose [≤ 120 days], but with little or no attention to which vaccines were used, what their initial standard dose[s] regimen are (1 or 2), and at this point in the pandemic, the number of booster doses received should be part of the analysis. In short, for publication in Vaccines, I would expect a more complex epidemiological analysis primarily of vaccine and vaccination variables with population and co-morbidity differences as secondary hypotheses or observations. The greater focus in this ms. on comorbidities is important but is better suited to other journals.